# Placental Changes and Neuropsychological Development in Children—A Systematic Review

**DOI:** 10.3390/cells12030435

**Published:** 2023-01-28

**Authors:** Maria Lodefalk, Felix Chelslín, Johanna Patriksson Karlsson, Stefan R. Hansson

**Affiliations:** 1Department of Pediatrics, Faculty of Medicine and Health, Örebro University, 701 82 Örebro, Sweden; 2University Health Care Research Centre, Faculty of Medicine and Health, Örebro University, 701 82 Örebro, Sweden; 3Department of Obstetrics and Gynecology, Institute of Clinical Sciences Lund, Lund University, 221 00 Lund, Sweden; 4Department of Obstetrics and Gynecology, Skåne University Hospital, 214 28 Malmö, Sweden

**Keywords:** autism spectrum disorders, childhood, developmental origins of health and disease, long-term outcome, neuropsychological development, pathological lesion, placenta, sexual dimorphism

## Abstract

Placental dysfunction may increase the offspring’s later-life disease risk. The objective of this systematic review was to describe associations between pathological placental changes and neuropsychological outcomes in children after the neonatal period. The inclusion criteria were human studies; original research; direct placental variables; neuropsychological outcomes; and analysis between their associations. The exclusion criterion was the offspring’s age—0–28 days or >19 years. The MEDLINE and EMBASE databases were last searched in May 2022. We utilized the ROBINS-I for the risk of bias assessment and performed a narrative synthesis. In total, 3252 studies were identified, out of which 16 were included (i.e., a total of 15,862 participants). Half of the studies were performed on children with neonatal complications, and 75% of the studies reported an association between a placental change and an outcome; however, following the completion of the funnel plots, a risk of publication bias was indicated. The largest study described a small association between placental size and a risk of psychiatric symptoms in boys only. Inconsistency between the studies limited the evidence in this review. In general, no strong evidence was found for an association between pathological placental changes and childhood neuropsychological outcomes after the neonatal period. However, the association between placental size and mental health in boys indicates a placental sexual dimorphism, thereby suggesting an increased vulnerability for male fetuses.

## 1. Introduction

Since the first report of an association between lower birth weight and a higher risk for cardiovascular death in men [1], the research interest in elucidating the relationship between fetal life and adult diseases has been considerable. The hypothesis of developmental origins of health and disease (DOHaD) initially suggested that fetuses and infants respond to limitations in nutritional supply, thereby causing lifelong alterations in different tissues. These changes were meant to predispose the offspring to an increased risk for a later occurrence of type 2 diabetes, hypertension, and coronary heart disease [2]. The potential impact of infant growth patterns in respect of later disease risk was also acknowledged. Subsequently, the hypothesis was broadened in order to encompass other exposures than simply poor nutritional supply, such as early life over-nutrition and prenatal exposure to stress or environmental chemicals [3,4,5]. However, despite numerous human observational and experimental animal studies [6], the evidence for an optimal growth pattern in infancy in order to reduce the disease burden in adulthood is still limited [7].

The placenta, a transient fetal organ with multiple functions, is essential for providing nutrients to the growing fetus [8]. It is crucial for maintaining the pregnancy and acts as the fetus’ lungs, gut, endocrine glands, liver, and kidneys. Thus, besides supplying nutrients, the placenta provides the fetus with oxygen, removes waste products, and produces a variety of hormones, cytokines, and neurotransmitters [9,10]. It is well documented that placental disorders may lead to poor fetal growth and low birth weight [11,12,13,14,15]. In line with this knowledge, some researchers claim that the placenta is the most important organ for mediating fetal programming and plays a significant role in the development of adult diseases [16,17].

Indeed, placental dysfunction may profoundly impair neonatal life secondary to, for example, preterm birth [18,19] or asphyxia [20], which in turn increase the risks of neonatal death, sepsis, intraventricular hemorrhages, and other neonatal complications [21]. On the other hand, the potential impact of placental disorders on childhood morbidity after and irrespective of the neonatal period, is less well known. Studies on the associations between placental changes and childhood disorders have mainly focused on neuropsychological impairments. However, certain studies have reported associations with atopic diseases [22] or tumors [23]. Comprehensive, but not systematic, reviews have explored potential associations between placenta and brain development in children [9,24], as well as between placenta and atopic diseases [25,26]. Additionally, recently published reviews have described placental inflammation and its potential impact on the offspring’s developing immune system [27]. The potential role of the placenta in the development of pediatric attention deficit hyperactivity disorder (ADHD) and autism spectrum disorders (ASD) has also been discussed [28]. The underlying mechanisms in respect of the associations between prenatal exposures, placental changes, and the neuropsychological development of the offspring have mainly been investigated utilizing animal models and comprise several, different biological pathways. Maternal immune activation of transgenic mice was shown to increase placental interleukin (IL) 6 signaling and fetal brain inflammation, as well as to also induce cerebellar pathologies and behavioral abnormalities in the offspring [29]. Maternal exposure to the pro-inflammatory cytokine IL-1β led to hypoplastic placentas in mice, an infiltration of T cells in the placenta, and cortical atrophy in the fetal brain [30]. Male mice, but not female, who were depleted in respect of the placental insulin receptor gene showed signs of impaired brain cortical development and difficulties in sensorimotor gating [31]. This, and other findings of sex differences, have given rise to an increasing interest in placental sexual dimorphism [32], which may explain some of the sex differences observed in regard to adult disease risk.

Placenta-specific depletion of the insulin-like growth factor 2 gene (*Igf2*) in transgenic mice resulted in increased offspring anxiety [33]. The ligation of the uteroplacental vessel in rabbits led to placental under-perfusion, increased rates of stillbirths, reduced birth weight, and impaired neurobehavior of the offspring [34]. A maternal deficiency of long-chain polyunsaturated fatty acids was associated with suboptimal visual acuity and reduced levels of docosahexanoic acid (DHA) in the cerebral cortex and retina of newborn rhesus monkey offspring [35]. Learning problems, ADHD related behavior, and upregulation of pro-inflammatory gene expression in central nervous tissue were found in the offspring to DHA-deficient rats [28].

In humans, differential methylation of the placental genome was found in offspring that later developed ASD [36]. Variations in the methylation of placental genes involved in cortisol signaling (*NR3C1* and *HSD11B2*) were linked to variations in human offspring’s early-life behavior, such as: crying, attention, need for emotional soothing, movement quality, excitability, and habituation [9]. Variations in methylation of the placental serotonin receptor gene (*HTR2A*), placental leptin gene (*LEP*), imprinted genes, and differential expression of placental microRNAs have also been associated with variations in infant neurobehavior [9]. Increased placental vascular resistance—measured as abnormal fetal and umbilical blood flow, as well as poor fetal growth—were found to be associated with a reduction in the retinal optic nerve area in human adolescents [37].

As outlined above, several hormones (insulin, insulin-like growth factor-2, cortisol, and leptin) were found to be involved in the pathogenesis of aberrant offspring neuropsychological development via placental changes. In addition to these hormones, thyroid hormones are also important for normal trophoblast function, fetal brain development, and childhood cognitive function [38,39,40]. Indeed, profound maternal hormonal changes occur during pregnancy [41], mainly elicited by the placenta in order to enable pregnancy maintenance and fetal growth. Evidence suggests that deviations in these hormonal adaptations may influence fetal programming and stimulate the development of the adult metabolic syndrome [42]. Additional evidence of the importance of hormones for fetal programming and offspring morbidity is found in the fact that estradiol appears to protect, in rodents born after intra-uterine growth restriction (IUGR) [43], from postpubertal hypertension.

Several parts of the fetal brain have been reported to be affected by prenatal exposures, such as the cerebellum, cerebral cortex, retina, and optic nerve, as mentioned above. In addition, white matter damage was shown in the neonatal brain after prenatal methamphetamine exposure [44], prenatal Zika virus exposure [45] and after asphyxia in especially preterm-born infants [46]. Further, cerebellar or vermis atrophy were found in children with developmental problems, such as difficulties in verbal learning and memory performance, after prenatal alcohol exposure or extreme prematurity [47].

As far as we know, only one systematic review has been published regarding the associations of placental changes with childhood outcomes [48]; however, that study only searched one database, restricted the placental change to placental pathological lesions, and was published some years ago.

In order to explore the potential impact of placental changes regarding childhood morbidity, this project aimed to perform a systematic review of the literature on associations between any placental change and childhood outcome beyond the neonatal period, whereby neuropsychological development, atopic diseases, and tumors was specifically focused upon. Child outcome was limited to these three most studied areas in order to avoid applying a too broad approach. As we aimed to elucidate potential evidence for an association of placental changes per se with pediatric conditions, neonatal morbidity was considered an important confounding factor.

## 2. Material and Methods

### 2.1. Eligibility Criteria

This review was performed and is reported in accordance with PRISMA guidelines [49]. In addition, it was registered with the International Prospective Register of Systematic Reviews (PROSPERO), registration number CRD42021276562. In order to be included in the review, each study had to describe a placental variable, which had to be investigated in relation to a deviation of the child that occurred after the neonatal period but before an age of 19 years. The deviation of the child was limited to tumors, atopic diseases, and neuropsychological development. However, even though infantile and congenital hemangiomas are types of tumors, they were not eligible as they often present in the neonatal period. Only studies written in English, as well as original research articles or case reports were eligible. Further exclusion criteria were publications before 1 January 2000 and animal studies.

All types of placental variables were eligible, for example, differential gene expression, weight, as well as histopathological or culture findings of placental tissue. However, isolated investigations of only the fetal membranes, umbilical cord, or amniotic fluid were not sufficient for inclusion as we focused on issues regarding the placental disc in this study. The placental variable also needed to have been investigated directly in the study by using placental samples. Thus, registry-based studies on, for example, preeclampsia without any investigation of placental tissue were not eligible; this is even though preeclampsia originates from abnormal placentation [50]. Further, studies reporting only indirect measures of a placental pathological change—such as aberrant Doppler wave signals in fetal cord vessels, which indicated increased placental vasculature resistance—were not eligible either. Studies only reporting a combined placental variable, such as placental weight/birth weight, were not eligible either, in order to enable the studying of placental aberrancies per se. In addition, studies investigating placental abruption were not eligible, given the considerable impact of this placental complication on neonatal life [51], as well as due to the review’s objective of studying the outcomes of offspring after and irrespective of the neonatal period. Finally, studies regarding twin–twin transfusion syndrome were not included due to the difficulties in differentiating the potential effects of the placental disorder from the effects of the aberrant blood flow to the fetuses, besides the impact of the syndrome on the neonatal period [52]. The neonatal period was defined as the first 28 days of life, corrected for prematurity [53].

### 2.2. Information Sources and Search Strategy

Two experienced librarians built search blocks that were tested, refined, and finally used. The search strategy included medical subject headings (i.e., MeSH) terms, free text words, and study-type restrictions (see Appendix A). The EMBASE and MEDLINE databases were searched for eligible studies in June 2021 and then repeatedly on 13 May 2022. Duplicates were removed. Relevant review articles and the bibliographies of included studies were also searched for additional studies.

### 2.3. Selection Process

Two reviewers (ML and JPK) independently screened all titles and abstracts received from the first database search. Next, the potentially relevant studies that were identified were read in full text by the same two researchers, independently, in order to determine eligibility. Disagreements were resolved by discussions, which led to consensus. After the second search, a further two reviewers (ML and FC) independently screened all titles and abstracts, as well as independently read all identified potentially relevant studies in full text. Again, disagreements were resolved by discussions. Finally, one reviewer (ML) screened the bibliographies of relevant review articles and the included studies for additional potentially relevant studies.

### 2.4. Data Collection Process and Items Extracted

One reviewer (ML) collected data manually from each included study, and another (FC) checked their correctness. Data were extracted to a custom-made standardized form in Microsoft Excel. The extracted data consisted of information regarding study characteristics, such as authors, publication year, context, aim, study population, and primary outcome. It also consisted of information regarding exposure (i.e., placental change, such as size, pathological lesion, or gene expression); offspring outcome and age; how an association between exposure and outcome was investigated, as well as the results of such analysis; the handling of confounding factors; and funding sources. All associations between relevant exposure and outcome in each included study were extracted. Any assumed data did not replace missing or unclear information in the included studies. No extra information, apart from published data, were collected from the authors of the included studies.

### 2.5. Study Risk of Bias Assessment

The risk of bias (RoB) assessment was independently performed by two reviewers (ML and FC). Furthermore, it was based on the focus of this review and not on the aims of the individual studies. The RoB assessment was performed using the preferred Cochrane tool for non-randomized studies (ROBINS-I) [54] adjusted for exposure (i.e., the placental variable) instead of intervention. Fetal sex, gestational age (GA) at birth, and neonatal morbidity were identified as important confounding factors for all or most studies prior to the RoB assessment. Therefore, they were considered in all assessments. Furthermore, any study that did not control for all these three variables were judged as possessing a serious RoB in the confounding domain, as outlined in the instructions to ROBINS-I [55]. These covariates were chosen as they have been found to be associated with both exposure and the outcome of interest for this review [18,19,20,32,56,57,58,59]. In order to facilitate uniform assessment among studies, we considered a failure to select participants of <10% of the eligible population or missing outcome data of <10% as a low RoB in respect of the domains on selection of participants and missing data, respectively. In addition, 10–20% were recorded as a moderate RoB and >20% as a serious RoB. Disagreements between assessors were solved by discussions, such that consensus was reached for every domain and study.

### 2.6. Effect Measures

The effect measure used in each included study—for example, an odds ratio with a 95% confidence interval (CI)—was used when presenting the results of individual studies as well as during the synthesis of the included studies, as appropriate.

### 2.7. Synthesis Methods

Given the comprehensive and broad approach in the project, a meta-analysis generating a single effect measure by combining the results from the individual studies was not possible. Instead, we constructed a narrative synthesis. Results from studies with a similar design were visualized in a forest plot by using a Stata version 17.0 (Stata Corp., College Station, TX, USA). Descriptive and comparative statistical analyses (i.e., Fisher’s exact test for categorical variables, the Mann–Whitney U test for continuous variables, and Spearman’s rank test for correlations) were performed in an IBM SPSS Statistics version 27 (IBM Corp., Armonk, NY, USA).

### 2.8. Reporting Bias Assessment

Funnel plots were performed in order to assess the potential risk of missing results in the synthesis arising from publication bias [60]. Egger’s test was not performed, as recommended, given the small number of studies using the same type of effect measure [60].

## 3. Results

### 3.1. Study Selection

The initial search of the EMBASE and MEDLINE databases identified 2836 studies (see Appendix A). After screening their titles and abstracts, 443 studies were selected as potentially relevant by at least one of the reviewers and were read in full text. Of them, 127 fulfilled the inclusion criteria for the review. In the second search of the databases, 416 new studies were identified (Appendix A). Again, after the title and abstract screening, 50 were judged as potentially relevant by at least one reviewer and were read in full text. Of them, 15 fulfilled the inclusion criteria. Thus, 142 studies were included at this stage (Figure 1). Of them, 101 studies investigated a neuropsychological outcome; 15 investigated an atopic outcome; 6 investigated tumors including hemangiomas; 2 studies investigated both a neuropsychological outcome and atopy; and 18 examined other outcomes. Due to the large number of included studies, we decided to include only studies with a neuropsychological outcome. Therefore, the 103 studies with such an outcome were assessed for RoB (Figure 1). The studies investigating atopic diseases will be included in a future systematic review.

During the RoB assessment, 10 studies were excluded due to not fulfilling the inclusion criteria (six studies lacked relevant exposure, and four lacked relevant outcomes). Due to still possessing a large number of included studies and the known difficulties to draw firm conclusions from studies with poorer methods, we chose to include the studies with the lowest RoB only. This was defined as either low or moderate overall RoB or the following combination: serious RoB in the confounding domain only; moderate RoB in the selected reporting domain only; and low RoB in all other domains (i.e., selection of participants, classification of intervention (exposure), missing data, and measurement of the outcome). The RoB assessments of the excluded studies can be found in Appendix A. In total, 16 studies fulfilled the RoB criterion and were finally included in the review, which corresponded, in full, to 15,862 participants (Figure 1). Three studies [61,62,63] reported on childhood outcomes that were measured on several occasions. Still, only the results from the first evaluation were included in the review due to a high degree of missing data at later evaluations. The search in bibliographies of relevant review articles and included studies did not result in any additional studies to include.

### 3.2. Study Characteristics and Their Risks of Bias

The characteristics of each included study are shown in Table 1. The RoB assessments focusing on the exposure and outcome of interest for this review, and not on the aims of the individual studies, are shown in Table 2.

### 3.3. Results of Individual Studies

Ten studies investigated one or several histopathological placental lesions, including histological chorioamnionitis (HC); chronic villitis; maternal vascular malperfusion (MVM); fetal thrombotic vasculopathy; meconium staining; and degree of maturity [64,65,66,67,68,69,70,71,72,73]. Three studies examined placental gene expression [61,74,75]. Moreover, three studies investigated CpG methylation of promotor sites to genes involved in the hypothalamic–pituitary–adrenal (HPA) axis [76], xenoestrogen burden [62], and placental size [63], respectively. The most studied childhood outcome was psychomotor developmental delay, including neurodevelopmental impairments [62,68,69,70,71,72,73,76], followed by ASD or autistic traits [61,64,65,66,67] (Table 3). In half of the studies, the study population consisted of children with a complicated neonatal period, either due to preterm birth [64,67,68,69,70,71,76] or neonatal encephalopathy (NE) secondary to hypoxic-ischemic events [72]. The other half of the studies investigated subjects from the general population, i.e., children born full-term. This was even though, in one study, some of the children had been exposed prenatally to maternal gestational diabetes mellitus (GDM) [61]. Moreover, in another study, some of the children were exposed prenatally to a hurricane [74] (Table 1).

**Table 1 cells-12-00435-t001:** Characteristics of the included studies.

Source	Study Type	Setting	Primary Aim *To Study:*	Study Population	Primary Outcome
Hendson, 2011 [68]	Cohort	Canada Births 1997–2004	Survival and neuro- development outcome in VLBW infants exposed to HC	628 infants HC group: GA: 26.1 w +/− 0.1 BW: 899.3 g +/− 11.9 47% males Non-HC group: GA: 27.6 w +/− 1.0 BW: 958.7 g +/− 11.2 48% males	HC was associated with a lower risk of death after adjustment for perinatal variables, aOR = 0.44 (95% CI = 0.24–0.80)
Kaukola, 2005 [70]	Case-control	Finland Births 1998–2002	Doppler ultrasonographic parameters of fetal cardiovascular hemodynamics associated with poorer neurodevelopment	Cases: 7 infants with signs of PI and suboptimal outcome. GA: 29.1 w +/− 1.6 BW = 796 g +/− 376 Controls: 10 infants with signs of PI and normal outcome. GA: 29.4 w +/− 1.7 BW: 918 g +/− 249	6 of 9 Doppler ultrasonographic parameters differed between cases and controls
Khalife, 2012 [63]	Cohort	Finland, from the Northern Finland Birth Cohort. Births 1985–1986	Associations between placental size and psychopathology in childhood	8954 children 51% males GA: 39.4 w +/− 1.6 BW: 3575 g +/− 534	Placental size was associated with mental health outcomes in 8 years old boys
Limperopoulos, 2008 [67]	Cohort	Northern America Years of birth not stated	Prevalence of and risk factors for autistic features in children born preterm	91 children 60% males GA: 26 w (range 23–30 BW: 890 g (range 460–1490)	25% screened positive. Risk factors were GA, BW, chorioamnionitis, sex, and SNAP-II score
Meakin, 2018 [76]	Case-control	USA, drawn from the ELGAN study. Births 2002–2004	Associations between CpG methylation at HPA axis genes in placenta and cognitive impairment at 10 years of age	Cases: 70 children with moderate/severe cognitive impairment. Controls: 158 children with normal/low cognitive function. For all 228 children, GA: 25.7 w (range 23.0–27.6). 60% males	41 of 237 tested probes associated with moderate/severe impaired cognitive function
Mir, 2015 [72]	Cohort	USA Births 2006–2011	Associations between placental pathology and severity of NE and, in infants requiring hypothermia, neurodevelopmental outcome	120 neonates with NE 73 of them received hypothermia and were followed up. Their mean GA was 39 w +/− 2. BW: 3384 g +/− 607	9 infants receiving hypothermia died before 2 years of age. Placental pathologic findings were more common with increasing severity of the NE
Mir, 2021 [64]	Case-control	USA Births 2012–2015	Placental pathological lesions in children with ASD	Cases: 16 children with ASD GA: 26 w (25–75th centile: 25–29) Controls: 48 matched children GA: 26 w (26–29)	LGA placentas were more prevalent in the ASD group (31% vs. 4 %)
Nomura, 2021 [74]	Cohort	USA, enrolled 2010–2013 to the Stress in pregnancy study	Placental transcriptome in relation to natural disaster stress during pregnancy and child behavioral outcome	131 children. 38 of them were exposed prenatally to a storm. Mean GA: 39.2 w in both groups. Exposed group: BW: 3548 g +/− 577.5 52% males Unexposed group: BW: 3249 g +/− 649 44% males	221 genes were DE between exposed and unexposed placentas after FDR adjustment and when requiring a FC > 2
Soullane, 2022 [66]	Case-control	Canada Births 2000–2017	Associations between placental gross morphology and pathology and ASD	Cases: 107 children with ASD 78% males GA: 39.3 w (IQR: 38.6–40.1) BW: 3380 g (2995–3755) Controls: 526 matched children 52% males GA: 39.6 w (38.6–40.3) BW: 3370 g (3050–3675)	18% of children in both groups had at least one placental pathology. Gross morphology did not differ between groups
Spinillo, 2021 [71]	Cohort	Italy Births 2007–2015	Associations between placental pathological lesions, neonatal mortality and neuro- developmental outcome in VLBW infants	574 newborns Sex distribution not stated GA: 29.4 w (IQR 27–31.3) BW: 1100 g (IQR 854–1354)	Neonatal mortality: 14%. Four lesions associated with neonatal death
Straughen, 2017 [65]	Case-control	USA Births 2007–2014	Associations between placental pathology and ASD	Cases: 55 children with ASD 76% males GA: 37.4 w +/− 4.0 BW: 2996 g +/− 910.6 Controls: 199 matched children 75% males GA: 37.7 w +/− 3.7 BW: 3148.7 g +/− 833.6	Five of 18 lesions differed in prevalence between cases and controls
Thebault-Dagher, 2021 [75]	Case-control	Canada, enrolled 2010–2012 to the 3D cohort study	Placental expression of 14 genes in children with FS	Cases: 28 children with FS GA: 38.9 w +/− 1.6 BW: 3.3 kg +/− 0.4 Controls: 84 PSM children GA: 39.0 w +/− 1.2 BW: 3.5 kg +/− 0.4 64% males in both groups	Cases had DE of *NR3C1-β, SLC6A4, HTR2B, GJA1* and *TPJ1* in placenta
Torrance, 2010 [69]	Cohort	Netherlands Births 1997–2004	Prognosis and predictors of outcome in preterm IUGR children	180 children. 56% males GA: 30.2 w (26–33.9) BW: 875 g (440–1470)	Neonatal mortality: 9% Severe neonatal complications: 28%
Ueda, 2022 [73]	Cohort	Japan, drawn from the HBC study Births 2007–2011	Associations between placental pathology and neurodevelopment	258 children. 52% males GA: 38.4 w +/− 1.9 BW: 2793 g +/− 552	Three lesions were associated with lower scores and four lesions were associated with higher scores
Vilahur, 2014 [62]	Cohort	Spain, drawn from the INMA study Enrolled 2003–2008	Associations between prenatal exposure to xenoestrogens and neuropsychological development	489 children. 52% males GA: 40.0 w (IQR: 39.0–40.7)	TEXB-α tertiles were not associated with any outcome at 1–2 years of age
Zhu, 2021 [61]	Cohort	China, drawn from the MABC study Enrolled 2013–2014	Associations between prenatal exposure to GDM and autistic traits and ADHD symptoms, and whether placental cytokines play a mediating role	3260 children 13% prenatally exposed to GDM. GA: approx. 39 w 50% of the children that did not develop autistic traits nor ADHD was males	GDM exposure was associated with an increased risk of autistic traits but not ADHD symptoms

Abbreviations: ADHD, attention deficit hyperactivity disorder; aOR, adjusted odds ratio; ASD, autism spectrum disorders; BW, birth weight; DE, differential expression; ELGAN, extremely low gestational age newborn; FC, fold change; FDR, false discovery rate; FS, febrile seizures; GA, gestational age; GDM, gestational diabetes mellitus; HBC, Hamamatsu Birth Cohort for Mothers and Children; HC, histological chorioamnionitis; HPA, hypothalamic-pituitary-adrenal; INMA, Infancia y Medio Ambiente; IQR, interquartile range; IUGR, intrauterine growth restriction; LGA, large for gestational age; MABC, Ma’anshan Birth Cohort; NE, neonatal encephalopathy; PI, placental insufficiency; PSM = propensity score matched, SNAP, score of neonatal acute physiology; TEXB = total effective xenoestrogen burden; VLBW, very low birth weight.

**Table 2 cells-12-00435-t002:** Results from the risk of bias assessments of the included studies using ROBINS-I.

Study	Bias Due to or in:						
	Confounding	Selection of Participants	Classification of Exposure	Missing Data	Measurements of Outcomes	Selection of the Reported Result	Overall Bias Risk
Hendson, 2011 [68]	Serious	Low	Low	Low	Low	Moderate	Serious
Kaukola, 2005 [70]	Serious	Low	Low	Low	Low	Moderate	Serious
Khalife, 2012 [63]	Serious	Low	Low	Low	Low	Moderate	Serious
Limperopoulos, 2008 [67]	Moderate	Moderate	Low	Low	Low	Moderate	Moderate
Meakin, 2018 [76]	Serious	Low	Low	Low	Low	Moderate	Serious
Mir, 2015 [72]	Serious	Low	Low	Low	Low	Moderate	Serious
Mir, 2021 [64]	Serious	Low	Low	Low	Low	Moderate	Serious
Nomura, 2021 [74]	Serious	Low	Low	Low	Low	Moderate	Serious
Soullane, 2022 [66]	Serious	Low	Low	Low	Low	Moderate	Serious
Spinillo, 2021 [71]	Serious	Low	Low	Low	Low	Moderate	Serious
Straughen, 2017 [65]	Serious	Low	Low	Low	Low	Moderate	Serious
Thebault-Dagher, 2021 [75]	Moderate	Low	Low	Low	Low	Moderate	Moderate
Torrance, 2010 [69]	Moderate	Low	Low	Low	Low	Moderate	Moderate
Ueda, 2022 [73]	Serious	Low	Low	Low	Low	Moderate	Serious
Vilahur, 2014 [62]	Serious	Low	Low	Low	Low	Moderate	Serious
Zhu, 2021 [61]	Serious	Low	Low	Low	Low	Moderate	Serious

**Table 3 cells-12-00435-t003:** Results from the included studies sorted by placental change.

Source	Placental Change	Outcome	Age	Associations Found	Confounders Controlled for
Straughen, 2017 [65]	Histopathological findings (MVM, chronic inflammation, chronic uteroplacental vasculitis, dysmaturity, chronic obstructive vascular lesions, FVM, acute inflammation (=HC))	ASD	Not specified	aOR (95% CI) for Any acute inflammation: 3.14 (1.39–6.95) Acute inflammation in the chorionic plate vessels: 5.12 (2.02–12.96) Any chronic inflammation: 1.67 (0.74–3.75) Chronic uteroplacental vasculitis: 7.13 (1.17–43.38) MVM: 12.29 (1.37–110.69) Villous edema: 0.05 (0.0005–0.42)	Sex, GA, BW
Soullane, 2022 [66]	Histopathological findings (inflammation, vasculitis, degree of maturity, other abnormalities (meconium staining, ischemic infarct, single umbilical artery, chorioangioma, subchorionic fibrin deposition, Tenny-Parker changes)) Gross morphology	ASD	Not specified	Placental hyper maturity: ASD group 4.7% vs. control group 0.4% (*p* < 0.0001), but the degree of placental maturity was only assessed in 26% of cases and 19% of controls. No other differences were found. No differences were found between groups.	None
Mir, 2021 [64]	Histopathological findings (HC, VUE, MVM, fetal thrombotic vasculopathy, villous edema, SGA or LGA placentas)	ASD	Approx. 4 years	> 1 placental lesion: ASD group 69% vs. control group 33% (*p* = 0.01) Presence of LGA placenta + HC: 25% vs. 2% (*p* = 0.01) Presence of LGA placenta: 31% vs. 4% (*p* < 0.01) aOR for presence of multiple lesions: 6.5 (1.6–27.1)	Sex, GA, GDM, maternal age
Limperopoulos, 2008 [67]	Histopathological findings (HC, placental abruption, or infarction)	Autistic traits	22 months corrected age	aOR for HC: 16.240 (2.798–94.270)	Sex, GA, BW, SNAP-II score
Hendson, 2011 [68]	Histopathological finding (HC)	NDI ^1^	18 months corrected age	HC associated with MDI, adjusted regression coefficient: −3.93 (−7.52 to −0.33) HC was not associated with NDI after adjustments	PROM, intrapartum antibiotic exposure, antenatal corticosteroids, mode of delivery, GA, sex, singleton vs. multiple birth
Torrance, 2010 [69]	Histopathological findings (infarction, VUE)	Mental development	2 years	Chronic VUE associated with poor neurodevelopmental outcome, aOR: 3.19 (1.26–8.09)	Sex, GA, BW, BW <2.3 percentile, UA pH <7.0, primiparity, hypertensive disease, ROP, RDS
Kaukola, 2005 [70]	Histopathological findings (HC, perfusion defect)	Psychomotor development	1 year corrected age	No differences or associations found	None
Spinillo, 2021 [71]	Histopathological findings (HC, VUE, FVM, MVM, intravillous hemorrhage)	Psychomotor development	24 months corrected age	aOR for survival with normal neurodevelopmental outcome: MVM: 0.45 (0.22–0.92) FVM: 0.46 (0.22–0.45) HC: 0.75 (0.43–1.29) Loss of placental integrity: 0.73 (0.44–1.21) Intravillous hemorrhage: 0.38 (0.22–0.62) VUE: 1.54 (0.86–2.75)	Sex, GA, BW, type of delivery
Mir, 2015 [72]	Histopathological findings (HC, VUE, fetal vascular thromboocclusive disease, maternal placental underperfusion, retroplacental hemorrhage/infarction, SGA or LGA placentas)	Death or NDI ^2^	18–24 months	OR for death or NDI: Any major placental pathology: 3.50 (1.07–11.44) Patchy/diffuse chronic villitis: 9.29 (1.11–77.73) HC: 0.94 (0.36–2.47) HC with fetal response: 2.23 (0.87–5.73)	None
Ueda, 2022 [73]	Histopathological findings (11 lesions, see the column “Associations found”)	Psychomotor development	10–40 months	Total MSEL composite scores associated with: Accelerated villous maturation: -2.46 (−4.30 to −0.61) Thrombosis or intramural fibrin deposition: 3.07 (1.36 to 4.79) Avascular villi: 2.68 (0.15 to 5.21) Delayed villous maturation: −2.62 (−4.59 to -0.64 Fetal inflammatory response: 2.26 (0.25 to 4.28) MVM: −2.09 (−3.69 to −0.50) FVM: 3.41 (1.74 to 5.07) But not with decidual arteriopathy, HC, VUE, or deciduitis.	Sex, BW, parity
Nomura, 2021 [74]	Gene expression: Transcriptome	Behavior	4 years	28 of 221 DEG between prenatally storm-exposed and unexposed children were found to mediate child aggression and 5 DEG were found to mediate child anxiety	Maternal age, drug use, education, marital status, fetal sex, BW
Thebault-Dagher, 2021 [75]	Gene expression: 14 genes ^3^	FS and age at first seizure	Up to 2 years	FS group had (with medium effect size) increased expression of *SLC6A4, GJA1* and *TPJ1*, and decreased expression of *NR3C1-β* and *HTR2B* Increased *SLC6A4* expression predicted younger age at first FS (large effect size)	Sex, GA, labor prior to delivery, complications at birth
Zhu, 2021 [61]	Gene expression: cytokines ^4^	Autistic traits	18 months	None of the investigated mRNAs associated with autistic traits after FDR corrections	Maternal age, prepregnancy BMI, HDCP, place of residence, educational level, average monthly income, parity, smoking history, fetal sex, BW, delivery mode, GA, the other cytokine mRNA levels
Meakin, 2018 [76]	CpG methylation ^5^	Cognitive and executive function	10 years	41 probes showed methylation differences by cognitive functioning Highest OR was 1.876 (1.067–3.298) found for the TSS200 region near *NR3C1*	Race, public insurance, maternal education, fetal sex, GA
Vilahur, 2014 [62]	Total effective xenoestrogen burden	Mental and psychomotor development	11–22 months	No significant association between TEXB-α values and MDI or PDI scores was found	MDI: geographical area of origin, sex, parental social class, maternal age, CS, maternal height, GWG, passive smoking and log transformed TEXB-β values PDI: geographical area of origin, sex, maternal BMI, breastfeeding, parental social class, maternal height, marital status, and log transformed TEXB-β values
Khalife, 2012 [63]	Weight and surface	Psychiatric disturbance	8 years	For boys: aOR for placental weight Probable psychiatric disturbance: 1.14 (1.04–1.25) Antisocial disorder: 1.14 (1.03–1.27) Inattention-hyperactivity: 1.11 (1.00–1.24) Inattention: 1.11 (1.02–1.20) Hyperactivity: 1.12 (1.00–1.26) Neurotic disorder: 1.19 (0.99–1.42) For boys: aOR for surface area Probable psychiatric disturbance: 1.01 (1.00–1.03) Antisocial disorder: 1.02 (1.00–1.04) Inattention-hyperactivity: 1.02 (1.01–1.04) Inattention: 1.01 (1.00–1.03) Hyperactivity: 1.03 (1.01–1.05) Neurotic disorder: 1.00 (0.97–1.03) For girls: No associations were found	GA, BW, maternal age, family structure, education, social class, smoking during pregnancy, parity, pre-pregnancy BMI, GWG

^1^ NDI was defined as the presence of CP, MDI < 70 (mental delay), visual impairment, or sensorineural hearing loss. ^2^ NDI was defined as the presence of CP or a composite score of < 70 in any of the domains of cognition, language, or motor skills. ^3^ Fourteen genes linked to glucocorticoid or serotonin signaling or placental development/fetal growth were investigated (*CRH, NR3C-α, HSD11B1, NR3C-β, HSD11B2, TPH2, SLC6A4, MAO-A, HTR2A, HTR2B, GJA1, TPJ1, CSH1,* and *VEGF-A*). ^4^ Genes encoding the following inflammatory cytokines were investigated (IL-1β, IL-10, MCP-1, CRP, heme oxygenase 1 (HO-1), HIF-1α, glucose-regulated protein 78, TNF-α, IL-4, IL-6, IL-8, interferon-ɣ, CD206, CD68). ^5^ The degree of CpG methylation at the promotor sites of fourteen genes involved in the HPA axis was investigated using 237 probes. Abbreviations: ADHD, attention deficit hyperactivity disorder; aOR, adjusted odds ratio; ASD, autism spectrum disorders; BMI, body mass index; BW, birth weight; CI, confidence interval; CP, cerebral palsy; CS, cesarean section; DE, differential expression; DEG, differentially expressed genes; ELGAN, extremely low gestational age newborn; FC, fold change; FDR, false discovery rate; FS, febrile seizures; FVM, fetal vascular malperfusion; GA, gestational age; GDM, gestational diabetes mellitus; GWG, gestational weight gain; HBC, Hamamatsu Birth Cohort for Mothers and Children; HC, histologic chorioamnionitis; HDCP, hypertensive disorder complicating pregnancy; HPA, hypothalamic-pituitary-adrenal; INMA, Infancia y Medio Ambiente; IQR, interquartile range; IUGR, intrauterine growth restriction; LGA, large for gestational age; MABC, Ma’anshan Birth Cohort; M-CHAT, Modified Checklist for Autism in Toddlers; MDI, mental developmental index; MSEL, Mullen Scale of Early Learning; MVM, maternal vascular malperfusion; NDI, neurodevelopmental impairment; NE, neonatal encephalopathy; PDI, psychomotor developmental index; PI, placental insufficiency; PROM, premature rupture of membranes; PSM, propensity score matched; ROP, retinopathy of prematurity; RDS, respiratory distress syndrome; SNAP, Score of Neonatal Acute Physiology; TEXB, total effective xenoestrogen burden; UA, umbilical artery; VLBW, very low birth weight; VUE, villitis of unknown etiology.

The ages for outcome assessment ranged from 1 to 10 years (median age: 2.0 years). In two case–control studies, the ages at assessment were not clearly stated [65,66]. For children born preterm, the assessment age was corrected for prematurity, except for in the study by Torrance et al. [69], where it was not stated. Furthermore, a wide range of instruments was used for assessing the psychological outcomes (Table 4).

### 3.4. Results of Synthesis

Overall, 81% of the studies reported at least one statistically significant association between a placental change and a childhood neuropsychological outcome. However, one study examining nine different pathological lesions (i.e., HC, vasculitis, meconium staining, ischemic infarct, single umbilical artery, chorioangioma, subchorionic fibrin deposition, Tenny–Parker changes, and degree of maturity) and gross placental morphology found that only one variable, i.e., placental hyper maturity, was associated with the outcome [66]. However, that variable was only described in a minority of the cases and controls, and thus we considered that study as being without a significant association, which is also in accordance with the authors’ of the study [66]. Thus, 75% of the included studies reported an association between a placental variable and a neuropsychological outcome. The largest study found an association between placental size and ADHD, as well as other psychiatric symptoms in boys (n = 4596), but not in girls (n = 4358). However, the effect sizes were small, as they possessed a 95% CI ranging from 1.00–1.04 to 1.04–1.27 [63]. Small effect sizes were also seen for most probes in the study regarding placental CpG methylation and cognitive function [76]. Two studies found that some of the placental lesions were associated with better outcomes, while other lesions were associated with worse outcomes [65,73]. Specifically, villous edema lowered the risk of developing ASD, while any acute placental inflammation, acute inflammation in chorionic plate vessels, chronic uteroplacental vasculitis, and MVM increased the risk [65]. Furthermore, thrombosis or intramural fibrin deposition, avascular villi, fetal inflammatory response, and fetal vascular malperfusion (FVM) were associated with higher Mullen scale of early learning (MSEL) scores, while both accelerated and delayed villous maturation, as well as MVM were associated with lower MSEL scores, indicating slower psychomotor development [73] (Table 3).

Of the studies on children with a complicated neonatal period, 87.5% reported an association between a placental change and a childhood outcome, compared to 62.5% of the other studies (n.s.). All studies investigating children with neonatal morbidity examined placental pathological lesions except one, which studied CpG methylation at HPA axis genes [76]. The most prevalent outcome in these studies was psychomotor developmental delay (Table 3).

Histological chorioamnionitis was the most studied exposure considering all 16 studies. It was found to be associated with ASD or autistic traits in three [64,65,67] of four studies with such an outcome. The three studies describing an association included together 409 children, while the study not finding an association included 633 full-term born children [66]. HC was not associated with neurodevelopmental impairment or psychomotor development in five studies [68,70,71,72,73], but was negatively associated with a mental developmental index in one study [68]. In contrast, HC with a fetal inflammatory response was positively associated with higher scores of psychomotor development in another study [73]. Chronic villitis of unknown etiology also showed discordant results between studies [69,71,72,73].

Five studies (two with preterm born children) investigated ASD or autistic traits [61,64,65,66,67]. The sizes of the study populations ranged from 64 to 3260 children. In the smaller studies, a combination of LGA placenta and HC [64], or HC alone [67], was associated with ASD or autistic traits. However, in the larger studies, the results were more divergent. Straughen et al. (n = 254) demonstrated that certain placental lesions were positively associated with ASD, while another lesion—villous edema—was negatively associated with ASD [65]. Soullane et al. (n = 633) did not find any such association between placental histopathological lesions, or gross morphology, and ASD [66]. In addition, Zhu et al. (n = 3260) did not find any association between the placental gene expression of cytokines and autistic traits [61] (Table 3).

Psychomotor development was examined in relation to placental lesions in six studies (four with preterm born children [68,69,70,71] and one on children with severe NE [72]). The sizes of the study populations ranged from 17 to 628 children. In summary, all but the smallest study [70] showed that at least one of the studied lesions was associated with worse psychomotor development.

Two studies investigated whether differential placental gene expression was associated with both prenatal exposure and the outcome of the offspring. The exposures were a hurricane eliciting maternal stress during pregnancy [74] and GDM [61], respectively. Differential gene expression between stress-exposed and unexposed placentas was found to mediate differences in child aggression and anxiety [74], but the increased risk of autistic traits in children prenatally exposed to GDM was not found to be mediated by differential gene expression [61] (Table 3).

Six studies applied a similar study design [63,64,65,67,69,72] as they investigated placental size or pathological lesions in relation to worse neuropsychological outcomes and presented their findings as odds ratios. Their results are summarized in a forest plot showing mostly positive associations between the placental variable and the offspring outcome (Figure 2).

The included studies handled potential confounding factors differently. The number of variables controlled for ranged from none to 25 (Table 3), which is partly linked to study design. Only three studies controlled for a factor mirroring the neonatal period: the Score of Neonatal Acute Physiology-II [67], retinopathy of prematurity, respiratory distress syndrome [69], and complications at birth [75].

### 3.5. Reporting Biases

The results from the studies with a similar design (in total, 25 results) [63,64,65,67,69,72] were included in a funnel plot. The size of the male population (n = 4596) was used in the plot due to the fact that an association was only found in boys in the study by Khalife et al. [63]. The plot visualized that smaller studies reporting a positive or no association were missing, thereby indicating a risk of publication bias [60]. A funnel plot of only the main results from these studies showed a similar pattern. Further, a funnel plot without the large study examining placental size [63] did not result in the typical funnel shape that is consistent with a low risk of publication bias [60]. The odds ratio for the main finding from each of these six studies was also inversely associated with the size of the study population (rho = −0.829 and *p* = 0.042), thereby indicating a risk of publication bias or rather the so-called “small study effect” [60].

On the contrary, the sizes of the study populations did not differ statistically significant between the studies reporting an association between a placental variable and a childhood outcome and the studies not reporting such an association (204 children (min–max: 64–4596) vs. 561 children (min–max: 17–3260)).

## 4. Discussion

The broad approach in this systematic review was applied in order to elucidate the evidence for an association between any types of placental change per se and any types of neuropsychological outcomes in childhood. This resulted in the inclusion of studies with varying study designs. Placental changes ranged from size through pathological lesions and gene expression to estrogen burden. In addition, outcomes ranged from ASD through psychomotor developmental delay to febrile seizures. Furthermore, the study populations varied in sizes from 17 to 8954 children and included healthy, full-term born children, as well as preterm born children and children who had experienced severe NE. These variations hampered, as expected, the possibility of combining the results and performing a meta-analysis. Nevertheless, an overall picture can be discerned.

The largest study described a small association between increasing placental size and the prevalence of psychiatric symptoms, such as antisocial disorder and ADHD symptoms, in boys aged 8 years [63]. More specifically, when the placental weight increased by 100 g, the risk of probable psychiatric disturbance increased by 14% and the risk of inattention–hyperactivity increased by 11%. No corresponding findings were found in girls, which may partly depend on the lower frequency of ADHD symptoms found in females in general [77,78]. However, it might also be due to sexual dimorphism that has been observed in placentas, indicating that male fetuses may be more vulnerable to unfavorable conditions than female fetuses [32]. We have previously shown a sex difference in differentially DNA methylation in placentas in relation to preeclampsia and the exposure to ambient air pollution [79]. This is also in addition to the gene expression of the leptin receptor isoform b and in associations between inflammatory cytokines in placentas from severely obese women [80]. Sex differences originating in fetal life have also been shown to relate to cardiovascular performance [81], thereby potentially explaining some of the differences observed between men and women in cardiovascular disease risk. The frequencies of intellectual disability, visual impairment, and the referral to habilitation services have been shown to be higher in boys than in girls when they are both born extremely preterm [82]. This is in addition to the cognitive function at age 5–8 years, where boys were more affected than girls by IUGR and preterm birth [83]. These findings are in accordance with the suggested increased vulnerability in male fetuses. Further, a sex difference in the association between placental size and shape and hypertension in adults has been found [84]. Possible mechanisms for these findings may be linked to sex differences in placental growth and metabolism, which may be driven by the sex chromosomes and sex hormones [32]. Sex differences in placental expression of the glucocorticoid receptor and its isoforms, which are important actors in the stress response, may also be involved in the underlying mechanism. Normal physiological development of the placenta differs by sex in rats [85]; moreover, gene expression patterns differ by sex in both mouse and bovine blastocysts [86,87,88], as well as in early human placental progenitor cells [89]. In mid-gestation, the fetal and placental production of sex hormones increases, possibly influencing the further development of placental sexual dimorphism. In full-term human placentas, transcriptome studies have shown that more than 140 genes are differentially expressed by sex, mostly involving autosomes [90,91]. Animal and human studies have also shown sex differences in placental transport, immune and endocrine function in response to different prenatal exposures, mostly leading to inferior adaptive ability in males [32]. It has been postulated that the male placenta, in contrast to the female placenta, promotes growth rather than adaptation to environmental insults, which would program the male fetus to an increased susceptibility to later diseases [32]. Khalife et al. suggested that a larger placenta may reflect a compensatory mechanism to an unfavorable environment, such as a lack of nutrient supply, in order to ensure normal fetal growth. Nevertheless, the unfavorable environment may still have caused suboptimal fetal brain development, thereby increasing the risk of later psychiatric diseases, and male placentas may be more vulnerable than female placentas to a poor environment [63]. However, the effect sizes of the associations between placental size and ADHD symptoms in boys were small and only seen after the adjustments for several covariates [63]. No sensitivity analysis was presented, thus indicating that the results may be dependent on the choice of covariates included in the regressions. On the other hand, small effects by a single variable—such as placental weight—may be expected, as mental health disorders are multifactorial conditions [92].

Besides the findings in the large study by Khalife et al. [63], we conclude that there is not yet strong evidence for an association between a placental change per se and neuropsychological outcomes in children after the neonatal period. This is due to the fact that we found indications of publication bias, the “small-study effect” [60], and other limitations in this review. Publication bias is a well-known problem in science, referring to the tendency to more easily publish studies with a significant or a favorable result compared to other studies, thus limiting the possibility to draw balanced conclusions [93]. Discordant results were obtained for the outcome ASD and autistic traits in the included studies. Smaller studies found an association for HC [64,67], while larger studies did not find an association for their respective placental variable [61,66], which is indicative of the “small-study effect” [60,93]. However, differences between these studies in populations (pre- or full-term born children) and exposures (pathological lesions or gene expression) must also be acknowledged. Discordant results were also seen for two often investigated pathological lesions, HC and chronic villitis, thus indicating unclear evidence. Further, one study that mainly reported small effect sizes did not perform a correction for false discovery rate, even though as many as 237 probes were investigated [76]. In addition, results from two other studies [65,73] showed that some of their investigated pathological lesions were associated with a more favorable outcome, which is hard to explain. Other lesions were associated with a worse outcome, which is the expected finding, thereby indicating a risk of chance or random findings. Finally, only three studies [67,69,75] controlled for a variable reflecting neonatal morbidity, which we considered an important confounder as we aimed to investigate associations irrespective of the neonatal period. On the other hand, the need to correct for neonatal morbidity is lower in studies examining a general population when compared to studies examining children who experienced a complicated neonatal period.

Our conclusion agrees with previous results described in reviews on chorioamnionitis as they only found weak associations with neurodevelopmental outcomes [24,94]. However, it contrasts to a review on neuropsychiatric diseases that discussed different placental changes, including pathological lesions [95], but no systematic review on associations between placental changes and ASD has been published. Systematic reviews on pre- and early postnatal factors in association with ASD did not explicitly address placental changes [96,97,98]. Thus, our conclusion is not contradicted by other strong evidence. The GA at birth, fetal sex, and morbidity in the neonatal period—such as hypoglycemia, sepsis, and intraventricular hemorrhages—profoundly impact long-term outcomes [56,99,100,101,102], and such factors may be more important than placental disorders per se for later neuropsychological development in respect of the offspring. However, there is no doubt that the placenta is crucial for pregnancy maintenance and outcome [103], including fetal growth and development [11,12,13], and that placental diseases and dysfunctions can be deleterious to the pregnant woman and the fetus, thus affecting neonatal life [18,19,20]. Placenta-associated conditions, such as preeclampsia with and without fetal growth restriction, are also associated with an increased risk for long-term cardiovascular disease in the offspring [104].

Half of the included studies in this review investigated children who had had a complicated neonatal period, thus constituting a high-risk population for neurodevelopmental impairment. This high proportion may partly be due to clinical routines facilitating the study of these children. Many medical centers histopathologically examine all placentas from complicated births [68,72] and children who suffered from severe neonatal morbidity are closely followed up in clinical care, including standardized testing by a psychologist at certain ages [82,105]. A large number of prospective birth cohort studies have been launched during recent years for the study of healthy neonates, as well as in respect of the influence of different exposures in early life on their later health and disease risks. They include, among others, the Rhode Island child health study [106], the Generation R study [107], and the ENVIRONAGE birth cohort study [108]. Some of these birth cohorts were eligible for this review (see Table 1), but others were not as they, for example, examined the outcome only shortly after birth [106].

A limitation of the included studies in this review was found in the variation in respect of the description of the placental pathological lesions. This was probably due to standardizations that developed over time. Earlier studies referred to work by Redline et al. [109] and later studies referred to the Amsterdam Workshop Group Consensus Statement for diagnosing placental lesions [110]. Further, psychological outcomes were evaluated using at least 12 different instruments (see Table 4), partly due to varying ages at assessment and different aspects being evaluated. Still, a more standardized way to evaluate psychological functions in childhood in order to facilitate comparisons across studies is required. Whether there is a causal relationship between an exposure and an outcome cannot be answered by observational studies [111]. Such studies can only find associations, which can be influenced by confounding factors. There is always a potential for residual and unmeasured confounding, which can affect the effect size [112]. However, randomized, controlled, interventional studies, which can show causal effects, can hardly be performed due to ethical reasons when studying associations between placental changes and childhood neuropsychological outcomes. The included studies in this review controlled for confounders differently, but the variables most often controlled for were GA at birth, fetal sex, and birth weight, which is appropriate. However, the sex distribution differed between ASD cases and healthy controls in one study, as expected [56], but that difference was not controlled for [66], which possibly influenced the results.

The broad approach applied in this review enabled the identification of studies with a wide range of exposures and outcomes, which made it possible to comprehensively illustrate the potential impact of the placenta on childhood neuropsychological functions. Other strengths of this review were found in the restriction to only include studies with a direct measure of a placental parameter; the careful distinction made between clinical and HC; the close adherence to the PRISMA guidelines [49] including a pre-registered protocol; and the exclusion of studies with higher RoB. However, our review also possesses limitations. The broad approach limited the ability to study specific biological pathways. On the other hand, our aim was not to study a specific insult or biological mechanism, instead we aimed at finding all studies regarding associations between any placental change and childhood neuropsychological outcomes in order to scrutinize the evidence for such an association. If large and well-performed studies on a specific placental change were published, we would have identified them, thus limiting the potential negative impact of the broad search strategy on our conclusions. Another limitation was the restriction to include changes found in the placental disc only, potentially missing interesting associations linked to the fetal membranes or the umbilical cord. The RoB assessments of case–control studies were somewhat difficult. ROBINS-I is an appropriate RoB assessment tool for non-randomized studies [54], however, it was constructed for cohort studies. Currently, there is no better tool for RoB assessment of case–control studies (pers. com. Cochrane Sweden). Further, as a direct measure of a placental variable was required for inclusion in this review, we could not include large registry-based studies. However, other systematic reviews have previously summarized findings from such studies [113,114]. Given that the average age for the included children were 2 years, and given the aim of the review, we did not study the development of cardiovascular disease or its risk factors, which are the major outcomes according to the DOHaD theory [2]. However, except for obesity, risk factors for cardiovascular disease are unusual findings in children [115].

Even though we did not find strong evidence in general for an association between placental changes and childhood neuropsychological outcomes, the care of pregnant women still needs to be of high quality and easily accessible, thereby minimizing avoidable and known risks for the woman, fetus, and child. Future original research on this topic should include a large number of children, thus enabling adequate control for confounding factors, performance of sensitivity analysis, and the studying of sex differences. They should also investigate several placental changes—such as pathological lesions, gross morphology, and gene expression—at the same time. Furthermore, they should investigate the child’s neuropsychological outcome at a young age in order to minimize the loss of follow-up data. Standardized approaches for the purposes of measuring exposure and outcomes should be developed and implemented. Future studies should also investigate potential associations between placental changes and risk factors for cardiovascular diseases in childhood and adolescence. The potential impact of placental sexual dimorphism on fetal, childhood, and adult life should also be further explored. The findings of an association between placental size and mental health in boys [63] is required to be replicated in other populations.

In conclusion, increasing placental size was found to be associated with increased prevalence of psychiatric symptoms in boys but not in girls. This may be due to placental sexual dimorphism increasing the vulnerability of male fetuses. Otherwise, no strong evidence for a general association between placental changes—such as pathological lesions or differential gene expression—and childhood neuropsychological development was found in this systematic review.

## Figures and Tables

**Figure 1 cells-12-00435-f001:**
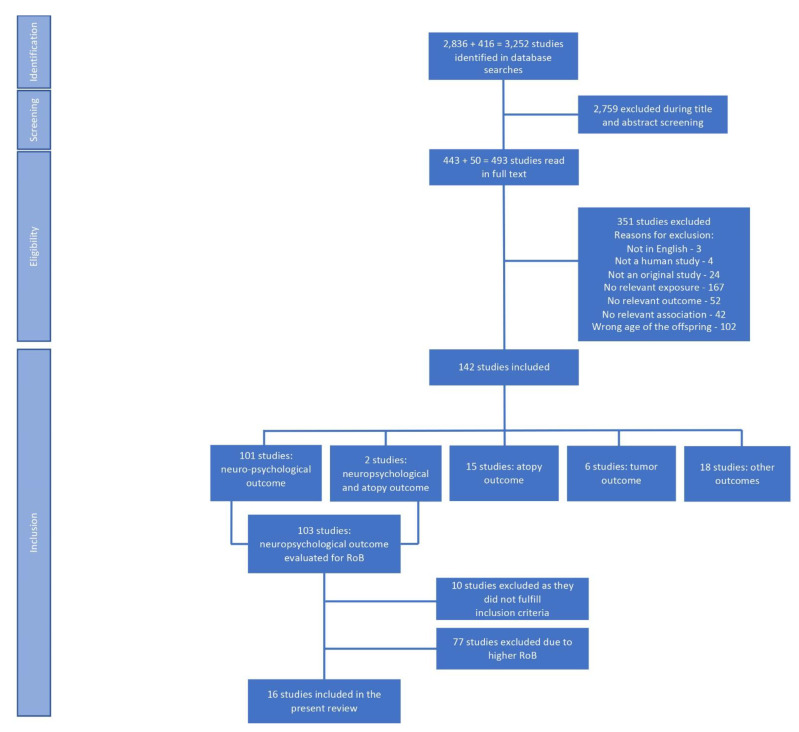
Flowchart of the process from identification to inclusion of the studies in the review.

**Figure 2 cells-12-00435-f002:**
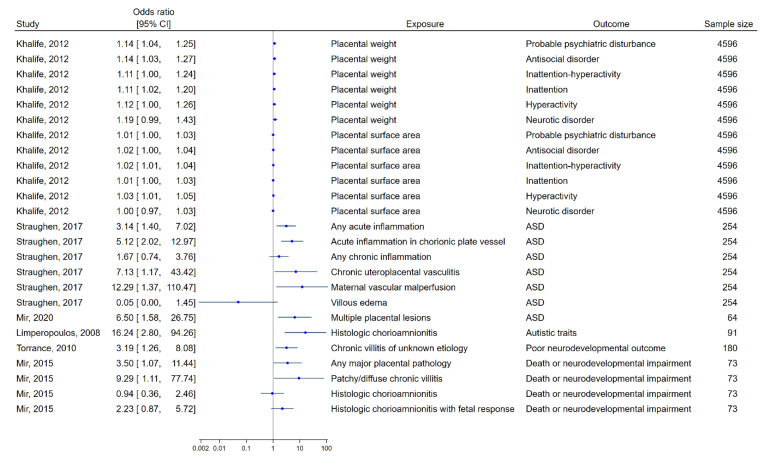
A forest plot detailing the risk (odds ratios (95% CI)) for different poor neuropsychological outcomes in childhood after the exposure of different placental changes (i.e., placental size or pathological lesions found at birth) as reported in six studies included in the review. The studies by Khalife et al. and Straughen et al. were mainly performed on full-term born children [63,65], while the other studies only were performed on children with a complicated neonatal period [64,67,69,75].

**Table 4 cells-12-00435-t004:** Instruments used in the included studies for the assessment of psychological outcomes in childhood.

Name of the Instrument	Outcome Measured	Studies Using the Instrument
Bayley Scales of Infant Development II or III	Mental developmental index	Hendson, 2011 [68], Torrance, 2010 [69], Vilahur, 2014 [62], Mir, 2015 [72], Mir, 2021 [64]
Griffiths Mental Development Scale	Developmental quotient	Torrance, 2010 [69], Kaukola, 2005 [70], Spinillo, 2021 [71]
Mullen Scales of Early Learning	Gross and fine motor, visual reception, receptive and expressive language	Ueda, 2022 [73]
School-Age Differential Ability Scales-II Verbal and Non-verbal reasoning scales	General cognitive ability (IQ), executive function, and working memory	Meakin, 2018 [76]
NEPSY-II	Executive function, auditory attention, set switching, concept generation, mental flexibility, and inhibition	Meakin, 2018 [76]
Vineland Adaptive Behavior Scale	Communication, daily living, socialization, and motor skills	Limperopoulos, 2008 [67]
Child Behavior Checklist	Behavioral and emotional problems	Limperopoulos, 2008 [67]
Behavior Assessment System for Children-2	Behavioral and emotional problems	Nomura, 2021 [74]
Rutter B2 scale	Psychiatric disturbance, especially ADHD symptoms	Khalife, 2012 [63]
CHAT-23	Autistic traits	Zhu, 2021 [61]
M-CHAT	Autistic traits	Limperopoulos, 2008 [67], Mir, 2021 [64]
Autism Diagnostics Observation Schedule-II	Assessment for diagnosing ASD	Mir, 2021 [64]
Childhood Autism Rating Scale-II	Assessment for diagnosing ASD	Mir, 2021 [64]

Abbreviations: ADHD, attention deficit hyperactivity disorder; ASD, autism spectrum disorders; CHAT, checklist for autism in toddlers; IQ, intelligence quotient; M-CHAT, Modified checklist for autism in toddlers; NEPSY, a developmental neuropsychological assessment.

## Data Availability

The data presented in this study are available in the study and the Appendix A.

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
