# Peer review of "Placental Changes and Neuropsychological Development in Children—A Systematic Review"

_cells, 2023, doi:10.3390/cells12030435_

Round 1

Reviewer 1 Report

This manuscript offers very interesting review of placental changes and neuropsychological development in children. The main finding is that male fetuses may be more vulnerable to placental disorders than female fetuses. The authors are encouraged to expand upon this finding and explain the possible mechanisms of placental sex differences. Why male placentas more vulnerable to poor environment?

The paper is a review, it seems that the author evaluated too much about the shortcomings of the included article. Objective generalize the paper’s opinion and reflect facts may be better.

I have many interests in the underlying mechanism of placental changes and neuropsychological development, I hope the authors can expand this section. Fox example, how did the placental changes affect fetal neuropsychological development?

Which part of the fetal brain was damaged that lead to children neuropsychological disease after prenatal exposure? I hope the authors can add this section.

Overall, the study design is solid and the paper is well written and results are presented clearly, I think it can be published after minor revision.

Reviewer 2 Report

The study is well written and very important to readers but some questions must be answered?

What about the inclusion and exclusion criteria?

Where is the history for case that author depend on ??? Do you have any selection basis??? Even many are suffered from affections in placenta... 

Introduction should be written in more clear form with respect to hormonal changes occured.   

Line 200-240 too much please summarize. 

Line 311-356 please rephrase those lines 

In general the conclusion should be totally rewritten... the discussion is good enough.... please pape needs English English editing 
